# FRL-SAGE: STACKELBERG GAME-THEORETIC DEFENSE AGAINST ADAPTIVE ADVERSARIES IN FEDERATED REINFORCEMENT LEARNING

## ABSTRACT

Federated Reinforcement Learning (FRL) enables multiple agents to collabora-tively train policies without sharing raw trajectories, but remains highly vulner-able to adversarial clients. Unlike supervised FL, FRL's sequential and policy-driven nature allows attackers to adapt strategies across rounds, while defenders must covertly reallocate protections in response. This evolving interaction natu-rally resembles a two-player strategic game, yet existing defenses assume static adversaries and fail to capture such dynamics. We propose FRL-SAGE (Stack-elberg Adversarial Game Equilibrium in Federated Reinforcement Learning), the first framework to formalize attacker–defender dynamics in FRL as a Stackelberg security game. The defender, acting as leader, commits to client-level protec-tions under a budget, while the attacker, as follower, best responds by selecting clients to compromise. We define asymmetric utilities: attacker utility is damage inflicted minus attack cost, while defender utility is the negative sum of residual damage and defense costs. The attacker's optimization reduces to a 0/1 knap-sack problem, solvable via dynamic programming or greedy heuristics, while the defender's bilevel planning is NP-hard but tractable through exact enumeration or scalable relaxation-based routines. To evaluate the framework concretely, we instantiate an adversary that uses gradient-noise injection and analyze four repre-sentative regimes, ranging from static single-client compromise to dynamic multi-client reshuffling with heterogeneous client importance. We theoretically establish equilibrium existence, prove computational hardness, and provide approximation guarantees for scalable solvers. Experiment on CartPole, a standard FRL testbed, illustrate that FRL-SAGE reduces attack-induced performance loss while oper-ating within realistic defense budgets, supporting its role as a principled game-theoretic foundation for proactive defense in adversarial FRL.

## 1 INTRODUCTION

Reinforcement learning (RL) has achieved remarkable success in domains such as gaming, robotics, and healthcare (Mnih et al., 2015; Silver et al., 2016; Li, 2017). Yet real-world adoption remains limited by poor sample efficiency: individual agents often lack sufficient trajectories to learn robust policies. A natural solution is to enable multiple agents, such as hospitals, robots, or vehicles, to share knowledge (Lowe et al., 2017; Foerster et al., 2016). However, raw trajectories often con-tain sensitive information and cannot be directly exchanged (e.g., clinical records or driving logs) (Sheller et al., 2020; Li et al., 2019) due to privacy regulations. Federated learning (FL) (McMahan et al., 2017; Konečný et al., 2016) addresses data scarcity and privacy concerns by enabling multiple clients to collaboratively train models without sharing raw data, improving privacy and practicality across domains such as mobile intelligence and healthcare (Sheller et al., 2020; Li et al., 2019). Building on this idea, Federated Reinforcement Learning (FRL) extends FL to sequential decision-making (Li et al., 2022; Kumar et al., 2020a), allowing agents to jointly learn global policies while keeping trajectories private. FRL is particularly attractive for applications such as autonomous driv-ing, personalization, IoT control, and network management, where both efficiency and privacy are critical (Yang et al., 2019; Li et al., 2022).

Despite these benefits, FRL is highly vulnerable to adversarial manipulation. In contrast to supervised FL, where poisoning often corrupts data or backdoors (Bagdasaryan et al., 2020; Xie et al., 2020), adversaries in FRL can interfere with training through a variety of manipulations. Even small perturbations, such as corruption of policy gradients, can cascade through aggregation and destabilize global performance (Huang et al., 2017; 2021; Sun et al., 2020). Moreover, attackers are adaptive: they reallocate budgets, reshuffle clients, and selectively target high-value participants across rounds. Static defenses, designed for fixed adversaries, fail under such evolving strategies (Li et al., 2022; Kumar et al., 2020b). In practice, defenders also adapt covertly, creating a cyclical dynamic akin to a two-player strategic game. Yet existing FRL defenses largely ignore this interaction, leaving robustness against adaptive adversaries an open challenge (Yang et al., 2019; Han et al., 2021).

To address this gap, we propose FRL-SAGE (Stackelberg Adversarial Game Equilibrium in FRL), the first framework to model adaptive attacker–defender dynamics in federated reinforcement learning. Our key insight is to cast the interaction as a Stackelberg game: the defender (leader) commits to a client-level protection strategy under a budget, while the attacker (follower) best-responds by selecting clients to compromise. We introduce damage weights $w_i$ to capture heterogeneous client importance, proving that the attacker's optimization reduces to a 0/1 knapsack problem, while the defender's bilevel planning is NP-hard. We provide exact, greedy, and relaxation-based algorithms with approximation guarantees, and instantiate our framework with gradient-noise injection attacks. Finally, we evaluate under four representative adversarial regimes (single vs. multi-client, static vs. reshuffling) across benchmark FRL environments, showing that FRL-SAGE effectively mitigates attack-induced degradation of global policies.

Our work makes below contributions on both the theoretical and experimental fronts:

- We propose FRL-SAGE, the first framework that explicitly models adaptive attacker-defender dynamics in federated reinforcement learning (FRL), capturing the sequential and policy-driven nature of gradient noise injection threats.

- We formalize attacker and defender utilities asymmetrically: the attacker maximizes damage minus cost, while the defender minimizes weighted damage plus defense cost. We introduce client-specific damage weights to capture heterogeneous influence, prove equilibrium existence, and show that the attacker's problem reduces to a $0/1$ knapsack, while the defender's planning is NP-hard.

- We instantiate gradient-noise injection attacks and introduce four representative adversarial settings: (i) single-client static, (ii) single-client periodic shuffling, (iii) multi-client static, and (iv) multi-client periodic shuffling, reflecting realistic adversarial strategies with varying difficulty for the defender.

- We theoretically establish the existence of Stackelberg equilibria, prove NP-hardness of the defender's optimization, and derive utility approximation bounds for scalable heuristic strategies. Experiments serve as sanity checks.

- We propose to solve small FL systems precisely via dynamic programming (attacker) and enumeration (defender), and design scalable bilevel routines (e.g., SLSQP) with greedy knapsack heuristics for large-scale deployments.

- We evaluate FRL-SAGE on benchmark FRL environments, demonstrating consistent robustness gains and higher defender utility compared to state-of-the-art baselines, even under strong adaptive attacks.

## 2 BACKGROUND

We study a synchronous cross-client federated reinforcement learning (FRL) setup where multiple clients collaborate to train a shared global policy without exchanging raw trajectories (McMahan et al., 2017; Konečný et al., 2016; Yang et al., 2019). FRL is relevant to settings such as distributed robotics, IoT control, and privacy-sensitive applications (Yang et al., 2019; Li et al., 2022), but here we focus only on the technical setup used in our framework. See Appendix A.1 for an extended discussion of related work.

## 2.1 Federated Reinforcement Learning Setup

Formally, we consider $K$ clients $C = \{1, \ldots, K\}$, each with a local Markov decision process (MDP) $M_i = (S, A, P_i, R_i, H, \gamma)$. All clients share state and action spaces $(S, A)$ but may differ in their local dynamics $(P_i, R_i)$. At round $t$, each client $i$ collects trajectories $\mathcal{D}_i$, computes policy gradients using REINFORCE (Williams, 1992), and updates local parameters $\theta_i^{(t)}$.

A central server aggregates the local updates using uniform federated averaging (McMahan et al., 2017):

$$\theta^{(t+1)} = \frac{1}{K} \sum_{i=1}^{K} \theta_i^{(t)}.$$

In our implementation, $K$ parallel workers each interact with independent environment instances (e.g., `CartPole-v1`), perform local gradient updates, and transmit them to the server for aggregation. This setup serves as the baseline FRL system on which we introduce adversarial dynamics and defense mechanisms.

We formalize federated reinforcement learning under adversarial gradient noise injection as a Stackelberg security game. The framework models how an attacker strategically selects clients to compromise through budget-constrained gradient corruption, and how a defender allocates limited protective resources in anticipation. The attacker injects scaled noise into policy gradients before aggregation, with attack intensity determined by client damage weights that reflect strategic importance in the federated system. We define all components of the system, establish equilibrium existence, analyze computational complexity, and connect theoretical constructs to sensitivity of global policy performance.

We adopt the general FRL framework introduced in Section 2, where $K$ clients collaboratively optimize a policy over a shared Markov decision process (MDP). Each client interacts with its own environment instance, collects trajectories, and updates its local parameters using policy gradient methods based on REINFORCE (Williams, 1992). At the end of each federated round $t$, the server aggregates the locally updated client parameters using uniform Federated Averaging (FedAvg) (McMahan et al., 2017), producing the new global model

$$\theta^{(t+1)} = \frac{1}{K} \sum_{i=1}^{K} \theta_i^{(t+1)},$$

where $\theta_i^{(t+1)}$ represents the parameters of client $i$ after local policy gradient updates. This iterative procedure enables clients to improve a shared global policy while keeping raw trajectory data local, thereby preserving privacy and reducing communication overhead.

We now formalize adversarial interactions in FRL using security games and Stackelberg games. This formulation captures the strategic decision-making of both attackers and defenders under resource constraints, and aligns with classical models in adversarial reasoning Tambe (2011); Conitzer & Sandholm (2006b).

**Security games formalization.** Let $D$ denote the defender and $A$ the attacker.

**Definition 1** (Security Game for FRL). *The security game is $\mathcal{G}_{\text{FRL}} = \langle \{D, A\}, S_D, S_A, U_D, U_A \rangle$ where:*

- $S_D, S_A \subseteq 2^{\mathcal{C}}$ *are strategy spaces (subsets of clients to protect/attack)*

- $U_D, U_A : S_D \times S_A \to \mathbb{R}$ *are utility functions*

**Definition 2** (Budget-Constrained Strategies). *Each client $i$ has defense cost $c_{d,i} \geq 0$ and attack cost $c_{a,i} \geq 0$. With budgets $B_D, B_A \geq 0$, the feasible strategy spaces are:*

$$F_D = \left\{ s_D \in S_D : \sum_{i \in s_D} c_{d,i} \leq B_D \right\}$$

$$F_A = \left\{ s_A \in S_A : \sum_{i \in s_A} c_{a,i} \leq B_A \right\}$$

**Stackelberg game formulation.** We model the defender-attacker interaction as a Stackelberg game where the defender acts as leader, committing to a protection strategy first, and the attacker responds optimally.

**Definition 3** (Stackelberg Game for FRL). *The Stackelberg game is $\mathcal{G}_{FRL}^{S} = \langle\{D, A\}, F_D, F_A, U_D, U_A\rangle$ with sequential play:*

    1. **Commitment:** *Defender commits to protection strategy $s_D \in F_D$*

    2. **Response:** *Attacker observes $s_D$ and chooses best-response:*

$$R_A(s_D) = \arg \max_{s_A \in F_A} U_A(s_D, s_A)$$

**Stackelberg equilibrium.** A strategy pair $(s_D^{\star}, s_A^{\star})$ constitutes a Stackelberg equilibrium if:

- $s_A^{\star} \in R_A(s_D^{\star})$ (optimal attacker response)
- $s_D^{\star} \in \arg \max_{s_D \in F_D} U_D(s_D, R_A(s_D))$ (optimal defender commitment)

**First-mover advantage.** For any Stackelberg equilibrium $(s_D, s_A)$ and corresponding Nash equilibrium $(n_D, n_A)$:

$$U_D(s_D, s_A) \geq U_D(n_D, n_A)$$

This result highlights the power of commitment in sequential games von Stengel & Zamir (2010b), capturing the defender's strategic advantage from committing to protection strategies before the attacker selects targets. Thus, in FRL, the defender benefits from committing protection resources before adversaries adapt their gradient perturbations. The full proof is in Appendix A.3.1.

## 2.2 THREAT MODEL

We initialize the threat model following Huang et al. (2021); Zhang et al. (2020), we assume that $m < K$ clients can behave adversarially during federated RL training Kumar et al. (2020a); Li et al. (2022). Each client $C_i$ possesses local trajectory data $D_i = \{(s_t, a_t, s_{t+1}, r_t)\}$ from its environment interactions. An adversarial client applies *gradient noise injection* by perturbing the computed policy gradients before aggregation:

$$\nabla_i' = \nabla_i + \xi_i \cdot (\max(\nabla_i) - \min(\nabla_i)) \cdot \frac{w_i}{w_{\text{ref}}}$$

where $\nabla_i$ represents the true policy gradients computed by client $C_i$, $\xi_i \sim \text{Uniform}(-1, 1)$ is uniform random noise sampled independently for each gradient element, $\max(\nabla_i)$ and $\min(\nabla_i)$ denote the maximum and minimum values within the gradient tensor $\nabla_i$, respectively, $(\max(\nabla_i) - \min(\nabla_i))$ represents the *gradient range* that adaptively scales the noise magnitude to the natural gradient scale of each parameter tensor, $w_i > 0$ is the *damage weight* representing the strategic importance of client $C_i$ in the federated system, and $w_{\text{ref}} > 0$ is a reference damage weight for normalization. The damage weight scaling $\frac{w_i}{w_{\text{ref}}}$ ensures that strategically more valuable clients (higher $w_i$) inject proportionally more disruptive noise, while the gradient range scaling ensures the attack adapts to each parameter's natural magnitude. This gradient corruption occurs during the local policy update phase, where Byzantine clients compute correct gradients but inject strategically-scaled noise before sending the corrupted gradients to the central server for federated averaging.

**Damage Weights.** Federated learning deployments naturally exhibit heterogeneity in client strategic importance due to factors such as data quality, representativeness, infrastructure criticality, and systemic impact Li et al. (2020); Wang et al. (2021). In healthcare federations, major hospital systems contribute more valuable and diverse data than individual clinics; in IoT networks, gateway nodes serve as critical aggregation points compared to edge sensors; in financial applications, major institutions have greater systemic importance than smaller participants Yang et al. (2019). Such heterogeneity creates natural incentives for adversaries to strategically target high-value clients rather than attacking randomly, yet existing federated learning security analyses typically assume uniform client importance Blanchard et al. (2017); Yin et al. (2018). To capture this realistic heterogeneity in our game-theoretic analysis, we model client strategic importance through damage weights $w_i$ that reflect the differential impact of compromising each client on overall system performance. Each $w_i$

represents the strategic importance of client $i$ in the federated system: clients with larger $w_i$ naturally amplify attack impact through the damage weight scaling $\frac{w_i}{w_{\text{ref}}}$ in the gradient noise injection, since compromising them yields greater influence on global policy performance. This heterogeneous damage weight assignment enables realistic game-theoretic analysis where defenders must prioritize protecting strategically valuable clients under budget constraints, reflecting the resource allocation challenges faced in actual federated learning deployments.

**Adversarial objectives and capabilities.** The attacker aims to degrade global policy performance by strategically compromising clients and maximizing the reduction in expected return $J(\pi_\theta)$. We consider a black-box adversary with no access to policy parameters $\theta$, local MDPs $\mathcal{M}_i$, or server aggregation logic. The attacker's sole capability is injecting gradient noise during local policy updates at compromised clients, corrupting the gradients $\nabla_i'$ before they are aggregated at the server.

**Attack strategy.** At each federated round, the attacker selects a subset $s_A \subseteq \mathcal{C}$ of clients to compromise, subject to budget constraints. The selection prioritizes clients with higher damage weights $w_i$, as compromising these clients yields greater impact on global performance. The attacker periodically reshuffles the compromised client depending on the costs every $t$ rounds, maintaining performance degradation while avoiding long-term anomaly patterns.

## 3 FRAMEWORK OF FRL-SAGE

### 3.1 STACKELBERG FRL GAME

We define $\mathcal{G}_{\text{FRL}} = \langle \{D, A\}, S_D, S_A, U_D, U_A \rangle$, where $D$ is the defender (server) and $A$ the attacker (malicious entity). Each client $i$ has a defense cost $c_{d,i} \geq 0$ and an attack cost $c_{a,i} \geq 0$. The defender chooses a protection set $s_D \subseteq \mathcal{C}$ with budget

$$\sum_{i \in s_D} c_{d,i} \leq B_D.$$

The attacker chooses an attack set $s_A \subseteq \mathcal{C}$ with budget

$$\sum_{i \in s_A} c_{a,i} \leq B_A.$$

The attacker gains damage from unprotected victims but pays attack costs:

$$U_A(s_D, s_A) = \sum_{i \in s_A \setminus s_D} w_i - \sum_{i \in s_A} c_{a,i}.$$

The defender pays defense costs and suffers residual damage:

$$U_D(s_D, s_A) = -\sum_{i \in s_D} c_{d,i} - \sum_{i \in s_A \setminus s_D} w_i.$$

**Assumptions:** (i) Perfect defense: we assume that protected clients cannot be successfully attacked (so a rational attacker never targets them) We assume perfect defense for analytical tractability, consistent with Stackelberg security games (Tambe, 2011). Extending to probabilistic defenses is left as future work. (ii) Perfect information: the attacker observes $s_D$ before acting. (iii) Rational players: both optimize their utilities.

**Definition 4** (Stackelberg Equilibrium). *A pair $(s_D^\star, s_A^\star)$ is a Stackelberg equilibrium if $s_A^\star \in \arg\max_{s_A} U_A(s_D^\star, s_A)$ and $s_D^\star \in \arg\max_{s_D} \min_{s_A \in R(s_D)} U_D(s_D, s_A)$, where $R(s_D)$ is the attacker's best-response set $R(s_D) = \arg\max_{s_A} U_A(s_D, s_A)$ subject to $\sum_{i \in s_A} c_{a,i} \leq B_A$.*

**Equilibrium existence:** At least one pure-strategy Stackelberg equilibrium exists (von Stengel & Zamir, 2010a).

*Proof:* We establish existence by showing that both strategy spaces are finite and utilities are well-defined, then applying standard results from finite games.

**Finiteness of strategy spaces.** The defender's strategy space $S_D$ consists of all subsets $s_D \subseteq \mathcal{C}$ satisfying the budget constraint $\sum_{i \in s_D} c_{d,i} \leq B_D$. Since costs $c_{d,i} > 0$ and the budget $B_D$ is finite,

only finitely many clients can be protected simultaneously. More precisely, $|s_D| \leq \lfloor B_D / \min_i c_{d,i} \rfloor$, which bounds $|S_D| \leq 2^{\lfloor B_D / \min_i c_{d,i} \rfloor}$. Similarly, the attacker's feasible strategy space $S_A = \{s_A \subseteq C : \sum_{i \in s_A} c_{a,i} \leq B_A\}$ is finite by the same argument.

**Well-defined utilities.** The utility functions $U_D(s_D, s_A)$ and $U_A(s_D, s_A)$ are real-valued and bounded on the finite domain $S_D \times S_A$, since all costs and damage weights are finite.

**Attacker's best response existence.** For any fixed defender strategy $s_D \in S_D$, the attacker solves $\max_{s_A \in S_A} U_A(s_D, s_A)$ over the finite set $S_A$. Since we are maximizing a real-valued function over a finite set, a maximizer $s_A^*(s_D) \in \arg\max_{s_A \in S_A} U_A(s_D, s_A)$ exists. When multiple maximizers exist, we select one using a deterministic tie-breaking rule (e.g., lexicographic ordering).

**Defender's optimization.** The defender solves $\max_{s_D \in S_D} \min_{s_A \in R(s_D)} U_D(s_D, s_A)$, where $R(s_D) = \arg\max_{s_A \in S_A} U_A(s_D, s_A)$ is the attacker's best-response correspondence. Since $S_D$ is finite and the objective function is real-valued, a maximizer $s_D^* \in S_D$ exists.

**Equilibrium construction.** Setting $s_A^* = s_A^*(s_D^*)$ yields the pair $(s_D^*, s_A^*)$ satisfying both equilibrium conditions: (i) $s_A^* \in R(s_D^*)$ (attacker best-responds), and (ii) $s_D^* \in \arg\max_{s_D \in S_D} \min_{s_A \in R(s_D)} U_D(s_D, s_A)$ (defender optimally commits). Therefore, $(s_D^*, s_A^*)$ constitutes a pure-strategy Stackelberg equilibrium.

## 3.2 ATTACKER'S PROBLEM: KNAPSACK REDUCTION

We establish that the attacker's optimization reduces to a standard 0/1 knapsack problem.

**Lemma 1** (Knapsack Reduction). *For fixed defender strategy $s_D$, the attacker's best response problem:*

$$\max_{s_A \subseteq C} \quad \sum_{i \in s_A \setminus s_D} w_i - \sum_{i \in s_A} c_{a,i}$$

$$s.t. \quad \sum_{i \in s_A} c_{a,i} \leq B_A$$

*is equivalent to the 0/1 knapsack problem.*

Under perfect defense, rational attackers only target undefended clients with positive utility. Let $I^+ = \{i \in C \setminus s_D : w_i - c_{a,i} > 0\}$ be the set of profitable attacks. The attacker's problem becomes:

$$\max_{s_A \subseteq I^+} \sum_{i \in s_A} (w_i - c_{a,i}) \quad \text{s.t.} \quad \sum_{i \in s_A} c_{a,i} \leq B_A$$

This is equivalent to a knapsack problem with items $I^+$, values $v_i = w_i - c_{a,i}$, weights $c_{a,i}$, and capacity $B_A$. The bijection between subset selection and binary variables preserves both objective and constraints. The detailed reduction is in Appendix A.3.3.

**Computational Complexity.** The attacker's problem can be solved exactly in $O(|I^+| \cdot B_A)$ time via dynamic programming, or approximated with a $\frac{1}{2}$-factor guarantee in $O(|I^+| \log |I^+|)$ time using greedy selection by efficiency ratio.

*Proof:* Standard knapsack DP uses table $DP[i, b] = \max$ utility from first $i$ items with budget $b$. The greedy algorithm sorts by efficiency $\rho_i = (w_i - c_{a,i})/c_{a,i}$ and achieves the classical $\frac{1}{2}$-approximation (Complete proof and detailed approximation analysis are given in Appendix).

This analysis gives us the theoretical foundation for efficient computation of attacker best responses, which is crucial for the defender's bilevel optimization problem in subsequent sections. See Appendix A.3.2 for the approximation proof.

## 3.3 DEFENDER'S BILEVEL OPTIMIZATION (DECISION–VECTOR FORM)

We encode the defender's protection strategy with a binary vector $x \in \{0, 1\}^K$, where $x_i = 1$ means client $i$ is fully protected and $x_i = 0$ means it is left exposed. The attacker chooses a binary vector $y \in \{0, 1\}^K$ indicating which clients to strike, under a budget $B_A$. Under the perfect-defense

assumption, a protected client yields no damage even if attacked; hence attacking client $i$ produces effective payoff $(w_i - c_{a,i})(1 - x_i)y_i$.

**Inner problem (attacker best response).** Given $x$, the attacker solves a 0/1 knapsack:

$$\mathcal{H}(x) = \max_{y \in \{0,1\}^K} \sum_{i=1}^{K} (w_i - c_{a,i})(1 - x_i)\, y_i \qquad (1)$$

$$\text{s.t.} \quad \sum_{i=1}^{K} c_{a,i}\, y_i \;\leq\; B_A.$$

Equivalently, letting $I^+(x) = \{\, i : (w_i - c_{a,i})(1 - x_i) > 0 \,\}$, $\mathcal{H}(x)$ is a knapsack on items $i \in I^+(x)$ with values $v_i = (w_i - c_{a,i})(1 - x_i)$, weights $c_{a,i}$, and capacity $B_A$.

**Outer problem (defender planning).** The defender pays protection costs and suffers residual damage $\mathcal{H}(x)$, while respecting its budget $B_D$:

$$\min_{x \in \{0,1\}^K} \quad f(x) = \sum_{i=1}^{K} c_{d,i}\, x_i \;+\; \mathcal{H}(x) \qquad (2)$$

$$\text{s.t.} \quad \sum_{i=1}^{K} c_{d,i}\, x_i \;\leq\; B_D.$$

Problems equation 1 & equation 2 precisely capture the Stackelberg structure: the defender (leader) chooses $x$, anticipating the attacker's best response value $\mathcal{H}(x)$.

*Proof sketch.* We reduce from the 0/1 knapsack decision problem. Given an instance with items $(c_i, v_i)$, capacity $C$, and target $V$, we ask whether there exists $z \in \{0,1\}^K$ such that $\sum_i c_i z_i \leq C$ and $\sum_i v_i z_i \geq V$.

*Construction.* For each item $i$, define a client with $c_{d,i} = c_i$, $c_{a,i} = 0$, and $w_i = v_i$. Set budgets $B_D = C$ and $B_A = 0$. Since $c_{a,i} = 0$ and $B_A = 0$, the attacker can attack all unprotected clients, so the best response is $y_i = 1$ whenever $x_i = 0$. Thus

$$\mathcal{H}(x) = \sum_{i=1}^{K} v_i(1 - x_i) = \sum_{i=1}^{K} v_i - \sum_{i=1}^{K} v_i x_i.$$

*Defender objective.* The defender's problem becomes

$$f(x) = \sum_{i=1}^{K} c_i x_i + \mathcal{H}(x) = \left( \sum_{i=1}^{K} v_i \right) + \sum_{i=1}^{K} (c_i - v_i)x_i,$$

subject to $\sum_i c_i x_i \leq C$. Since $\sum_i v_i$ is constant, minimizing $f(x)$ is equivalent to

$$\max_{x \in \{0,1\}^K} \sum_{i=1}^{K} v_i x_i \quad \text{s.t.} \sum_{i=1}^{K} c_i x_i \leq C,$$

which is exactly 0/1 Knapsack.

*Decision threshold.* Define $L = \sum_i v_i - V$. Then $f(x) \leq L \iff \sum_i v_i x_i \geq V$ under the same budget. Thus solving the defender's problem would solve 0/1 Knapsack. Therefore, the defender's optimization is NP-hard. See Appendix A.3.4 for the full reduction proof.

*Approximation oracle.* If the attacker oracle returns $\widehat{\mathcal{H}}(x) \geq \alpha\, \mathcal{H}(x)$ (e.g., $\alpha = \frac{1}{2}$ via greedy+best-item), then the scaled surrogate $\widetilde{f}_\alpha(x) = \sum_i c_{d,i} x_i + \frac{1}{\alpha}\widehat{\mathcal{H}}(x)$ upper-bounds $f(x)$ for all $x$, yielding a certified bound for the defender.

**Algorithmic Solutions.** Since exact solution is NP-hard, we develop three complementary approaches:

*Exact enumeration* for small instances ($K \leq 20$) guarantees global optimality by evaluating all $2^K$ feasible defense sets, with complexity $O(2^K \cdot |I^+| \cdot B_A)$ where the second term comes from the attacker oracle. See Appendix A.3.6 for the selection routine.

*Greedy protection* for medium instances sorts clients by protection efficiency $\rho_i = w_i/c_{d,i}$ and greedily selects high-efficiency clients within budget. This runs in $O(K \log K)$ time and provides intuitive approximation by prioritizing high damage-to-cost ratios.

*Bilevel optimization* for large systems ($K > 100$) uses SLSQP with continuous relaxation $x_i \in [0, 1]$ representing fractional client protection, then rounds to integer solutions. The attacker oracle provides gradients via finite differences, enabling scalable optimization.

The attacker oracle, fundamental to all approaches, solves the knapsack problem in Lemma 1 exactly via dynamic programming in $O(|I^+| \cdot B_A)$ time or approximately via greedy selection with $\frac{1}{2}$-approximation in $O(|I^+| \log |I^+|)$ time. A summary of computational complexity is provided in Appendix A.3.7.

Algorithm selection is automatic: exact for small instances, greedy for sparse budgets, and bilevel for dense budgets or large systems. Detailed pseudocode is provided in Appendix A.3.5.

## 3.4 RESULTS AND DISCUSSION

We use CartPole as our testbed because, despite its simplicity and noisiness, it is a controlled yet challenging stress test for adversarial robustness. Its quick convergence means small perturbations can destabilize learning, making it a strong litmus test for defenses. The high variance explains less consistent separation between strategies, but also underscores that if Stackelberg planning sustains advantages here, it is likely to generalize to more complex environments. All strategies are initialized under identical conditions to ensure fair comparison. Each of the 25 clients begins with independently seeded policy parameters and interacts with its own CartPole environment instance, eliminating bias from correlated trajectories. The defender and attacker budgets are expressed as ratios of total system cost, ensuring that resource allocations scale naturally with system size.

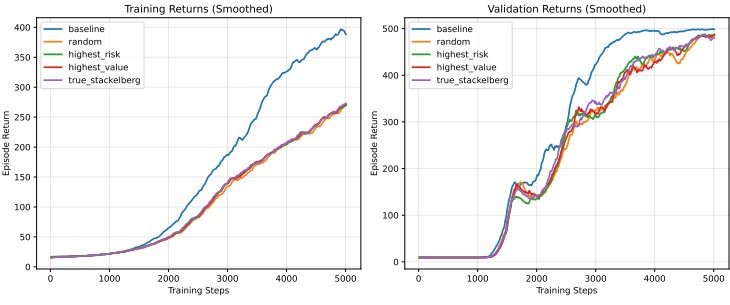

Figure 1: Single-client with periodic shuffling every 500 steps

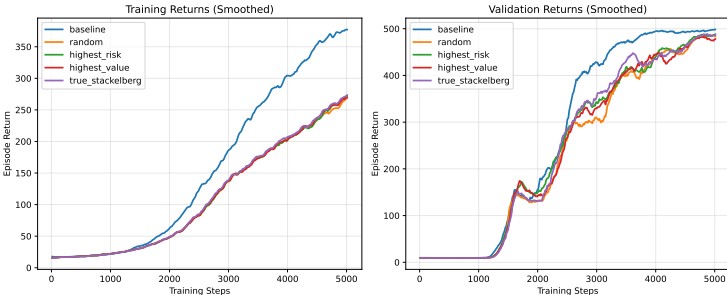

Figure 2: Single-client no periodic shuffling

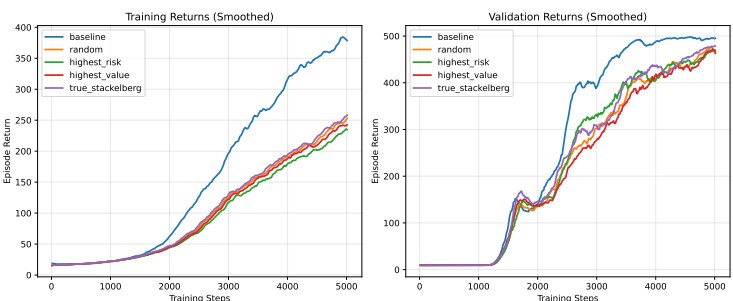

Figure 3: Multi-client with periodic shuffling every 500 steps.

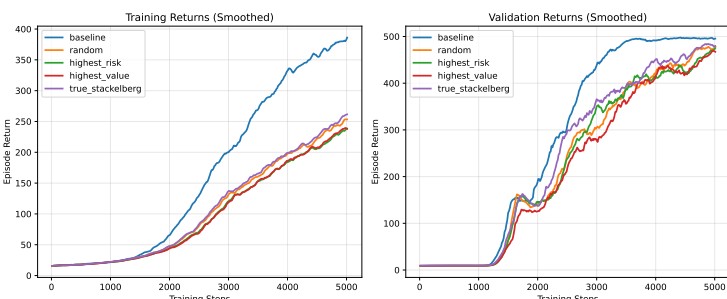

Figure 4: Multi-client no periodic shuffling

In the single-client 1 & 2, trajectories do not exhibit a dramatic or steadily widening gap between defenses. Because only one client is compromised at a time, the overall attack pressure is limited, and most strategies remain close together, with the defender's advantage appearing only in bursts. This reflects CartPole's rapid convergence and high variance, which amplify noise and mask small differences. In the multi-client setting 3 & 4, the gap between Stackelberg and the baselines becomes more visible, though still not overwhelming. We view this as consistent with the nature of our setup: CartPole is deliberately volatile, and adversarial FRL is resource-imbalanced by design. Importantly, even under such volatility, Stackelberg planning sustains a measurable edge over all heuristics, and the relative margin is expected to compound in richer environments where multi-client attacks exert stronger pressure and random variance plays a smaller role.

## 4 CONCLUSION

We introduced FRL-SAGE, the first Stackelberg game-theoretic defense framework for federated reinforcement learning under adaptive adversaries. Our formulation captures budgeted attacker–defender dynamics, provides theoretical guarantees, and yields practical approximation strategies. In CartPole, the defender's advantage emerges despite volatility and high variance, illustrating that even under deliberately challenging conditions, principled Stackelberg planning can secure measurable gains over heuristic defenses. We expect these benefits to become more pronounced in complex, real-world FRL environments where multi-client attacks exert stronger pressure and random variance plays a smaller role. At present, FRL-SAGE assumes perfect knowledge of client importance and binary defense decisions. In future work, we aim to (a) scale defender search using structure-aware pruning and learning-to-search heuristics, (b) extend binary defenses to continuous allocations that model partial safeguards such as throttling and redundancy, and (c) explore mixed-strategy Stackelberg equilibria that better capture real-world uncertainty.

## 5 REPRODUCIBILITY STATEMENT

We are committed to ensuring the reproducibility of our results. To this end, we provide all necessary details regarding the experimental setup, datasets, and evaluation within the manuscript. We also share source code and instructions to facilitate replication of our findings. Any external datasets or tools used are publicly available and properly referenced. Our aim is to enable independent researchers to reproduce and verify the results reported in this study.

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

## A APPENDIX

### A.1 RELATED WORK

A large body of work studies poisoning and backdoor threats against FL, showing that distributed training is highly sensitive to malicious client updates (Bagdasaryan et al., 2020; Xie et al., 2020), Fang et al. (2020); Baruch et al. (2019); Sun et al. (2021); Shejwalkar et al. (2022). Classical robust aggregators–Krum, Multi-Krum, coordinate-wise median/trimmed mean, and Bulyan–seek to limit Byzantine influence on the global update (Blanchard et al., 2017; Yin et al., 2018; El Mhamdi et al., 2018). Subsequent defenses incorporate trust modeling and server-side validation (e.g., FLTrust) (Cao et al., 2021). While these methods are useful against static or bounded adversaries, they give the defender no explicit way to *plan* under a strategic, budgeted attacker who reallocates resources across rounds. Our framework fills this gap by *optimizing* the defender's allocation in anticipation of the attacker's best response.

In single-agent RL, attacks can manipulate observations, actions, or rewards, often leading to large and persistent performance drops (Lin et al., 2017; Sun et al., 2020; Zhang et al., 2020; Han et al., 2021). Gradient corruption is particularly insidious: small perturbations to the policy gradients can steer updates in harmful directions with effects that compound over time (Zhang et al., 2020). However, most of these studies assume centralized training or focus on per-episode adversaries; they do not capture the cross-client, cross-round dynamics of federated settings.

FRL extends FL to sequential decision making, enabling distributed policy learning across clients without sharing trajectories (Yang et al., 2019; Li et al., 2022). Initial FRL works largely consider benign training, while a few recent efforts examine adversarial impact on FRL (Huang et al., 2021). Yet existing threat models are typically static or do not give the defender a principled mechanism to budget protections against adaptive, round-by-round attackers. FRL-SAGE directly addresses this by modeling the *sequential* attacker-defender interaction as a bilevel game and solving the defender's *planning* problem with explicit budget constraints.

Game-theoretic lenses have been used to study incentives and robustness in FL, including server-attacker minimax games and reputation-aware or incentive-compatible mechanisms (Jia et al., 2023; Li et al., 2024a; Hu et al., 2024; Zhang et al., 2022). Closer to our setting are leader-follower (Stackelberg) formulations, where commitment and response are asymmetric (Conitzer & Sandholm, 2006a; von Stengel & Zamir, 2010a; Tambe, 2011). The FL community has begun to explore Stackelberg-style reasoning for robustness and resource allocation, but predominantly in supervised FL or without explicit *budget-constrained* attacker models (Pan et al., 2023; Li et al., 2024b). Our work is distinct in (i) targeting *FRL* with gradient noise injection and episodic return; (ii) introducing client-specific *damage weights* that shape attacker value and defender prioritization; and (iii) proving that the attacker best response reduces to 0/1 knapsack with efficient oracles while the defender's problem is NP-hard. A closely related FL framework is the Stackelberg security game for supervised FL that models budgeted attacker/defender utilities and dynamic scenarios; we adapt this strategic structure to *reinforcement* learning with gradient noise injection, episodic return, and policy aggregation, and provide FRL-specific analysis and algorithms.

Complementary to game-theoretic planning are *certified* defenses that provide verifiable robustness guarantees, often via randomized smoothing, Lipschitz bounds, or stability analyses (Cohen et al., 2019; Lecuyer et al., 2019). In federated settings, certified or multi-level defenses (e.g., smoothing at client/server levels, hierarchical certificates) have been explored to bound worst-case degradation under bounded perturbations, as in MuCD's multi-level certified federation against poisoning (Liu et al., 2025). These methods deliver ex-post guarantees but typically assume norm-bounded noise models that may not capture an adaptive, budgeted adversary's *selection* of targets. Our approach is orthogonal and complementary: we plan ex-ante against *which* clients an adaptive adversary will attack under a budget (Stackelberg game), whereas certification focuses on *how much* an attack can change outcomes under a norm constraint.

## A.2 NOTATION TABLE

| Symbol | Meaning |
|---|---|
| $K$ | Number of clients |
| $C = \{1, \ldots, K\}$ | Set of clients |
| $s_D, s_A$ | Subsets of clients chosen for defense / attack |
| $c_{d,i}, c_{a,i}$ | Defense cost / attack cost for client $i$ |
| $B_D, B_A$ | Defender / attacker budget |
| $w_i$ | Damage weight (strategic importance of client $i$) |
| $w_{\text{ref}}$ | Reference weight for normalizing damage scaling |
| $U_D, U_A$ | Defender and attacker utility functions |
| $x_i, y_i$ | Binary defense / attack indicators for client $i$ |
| $H(x)$ | Attacker's best-response value (knapsack objective) |
| $t, T$ | Round index / total number of training rounds |

Table 1: Notation summary for FRL-SAGE.

## A.3 THEORETICAL ANALYSIS

This appendix provides detailed theoretical analysis, complete proofs, and additional experimental details that support the main results presented in the paper. We organize the material as follows: contains proofs of game-theoretic results including equilibrium existence and complexity analysis; provides detailed algorithms and implementation specifics.

### A.3.1 LEADER'S ADVANTAGE FOR GENERAL CASE

**Claim 1.** *For any Stackelberg equilibrium* $(s_D, s_A) \in E_S$ *and any Nash equilibrium* $(n_D, n_A) \in E_N$,
$$\mathcal{U}_\mathcal{D}(s_D, s_A) \geq \mathcal{U}_\mathcal{D}(n_D, n_A).$$

**1. Definitions.**

- $S_D, S_A$ are the finite action sets of Defender and attacker.
- In a Stackelberg equilibrium $(s_D, s_A)$, the follower selects

$$s_A \in \arg\max_{a \in S_A} \mathcal{U}_{\mathcal{A}}(s_D, a).$$

- In a Nash equilibrium $(n_D, n_A)$,

$$n_D \in \arg\max_{a \in S_D} \mathcal{U}_{\mathcal{D}}(a, n_A), \quad n_A \in \arg\max_{a \in S_A} \mathcal{U}_{\mathcal{A}}(n_D, a).$$

**2. Follower payoff comparison.** Since $n_A$ is a best-response to $n_D$ and $s_A$ is a best-response to $s_D$, we have

$$\mathcal{U}_{\mathcal{A}}(n_D, n_A) = \max_{a \in S_A} \mathcal{U}_{\mathcal{A}}(n_D, a)$$
$$\leq \max_{a \in S_A} \mathcal{U}_{\mathcal{A}}(s_D, a) = \mathcal{U}_{\mathcal{A}}(s_D, s_A).$$

**3. Leader payoff comparison.** Let $f^\star = \arg\max_{a \in S_A} \mathcal{U}_{\mathcal{A}}(n_D, a)$ be any best response of the follower to the leader's Nash action $n_D$. Because $(s_D, s_A)$ is a Stackelberg equilibrium, the leader chooses $s_D$ to maximize the utility given the follower's best response:

$$\mathcal{U}_{\mathcal{D}}(s_D, s_A) = \max_{x \in S_D} \mathcal{U}_{\mathcal{D}}\big(x, \mathrm{BR}_A(x)\big).$$

Hence

$$\mathcal{U}_{\mathcal{D}}(s_D, s_A) \geq \mathcal{U}_{\mathcal{D}}\big(n_D, \mathrm{BR}_A(n_D)\big)$$
$$= \mathcal{U}_{\mathcal{D}}(n_D, f^\star) \geq \mathcal{U}_{\mathcal{D}}(n_D, n_A)$$

where the final inequality uses that $n_A$ is also a best response to $n_D$.

### A.3.2  $\frac{1}{2}$-APPROXIMATION FOR THE DENSITY-ORDERING GREEDY ORACLE

**Claim 2.** *Consider the 0/1 knapsack instance induced by the attacker's best response, with item values $v_i = (w_i - c_{a,i})(1 - x_i)$, weights $c_{a,i}$, and capacity $B_A$. Assume $c_{a,i} > 0$ for all $i$ and restrict to profitable items $I^+ = \{ i : v_i > 0 \}$. Let $v_{\mathrm{int}}$ be the optimal integral value and let $v_{\mathrm{greedy}}$ be the value returned by the density–ordering greedy algorithm with best–single–item augmentation. Then*

$$v_{\mathrm{greedy}} \geq \tfrac{1}{2} v_{\mathrm{int}}.$$

*Proof.* Because we only consider $i \in I^+$, all values are non–negative and the problem reduces to a standard 0/1 knapsack instance. Let $v_{\max} := \max_{i \in I^+} v_i$ and let $v_{\mathrm{frac}}$ denote the optimal fractional knapsack value. Ordering items by density $\rho_i = v_i / c_{a,i}$, let $V_G$ be the total value of the maximal prefix that fits under capacity. If item $j$ is the first item that does not fit completely, then the fractional solution that takes a fraction $\lambda \in (0, 1)$ of item $j$ achieves value $V_G + \lambda v_j \geq v_{\mathrm{frac}}$. Thus

$$v_{\mathrm{frac}} \leq V_G + v_j \leq V_G + v_{\max},$$

so $V_G \geq v_{\mathrm{frac}} - v_{\max}$. The augmented greedy oracle returns

$$v_{\mathrm{greedy}} = \max\{ V_G, v_{\max} \} \geq \max\{ v_{\max}, v_{\mathrm{frac}} - v_{\max} \}. \tag{$\star$}$$

Now consider two cases.

(i) If $v_{\mathrm{int}} \leq 2 v_{\max}$, then $v_{\max} \geq \tfrac{1}{2} v_{\mathrm{int}}$, and by $(\star)$ we obtain $v_{\mathrm{greedy}} \geq v_{\max} \geq \tfrac{1}{2} v_{\mathrm{int}}$.

(ii) If $v_{\mathrm{int}} > 2 v_{\max}$, then $v_{\mathrm{frac}} \geq v_{\mathrm{int}}$ and hence $v_{\mathrm{frac}} - v_{\max} > \tfrac{1}{2} v_{\mathrm{int}}$. By $(\star)$, this implies $v_{\mathrm{greedy}} \geq \tfrac{1}{2} v_{\mathrm{int}}$.

Therefore in all cases $v_{\mathrm{greedy}} \geq \tfrac{1}{2} v_{\mathrm{int}}$. $\qquad\square$

This follows the classical analysis of the greedy density-ordering heuristic for 0/1 knapsack (Martello & Toth, 1990).

### A.3.3 KNAPSACK REDUCTION

For fixed defender strategy $s_D$, the attacker's best response problem:

$$\max_{s_A \subseteq C} \sum_{i \in s_A \setminus s_D} w_i - \sum_{i \in s_A} c_{a,i}$$

$$\text{s.t.} \quad \sum_{i \in s_A} c_{a,i} \leq B_A$$

is equivalent to the 0/1 knapsack problem. We show the attacker can restrict attention to $I^+$ without loss of optimality.

For any $j \in s_D$, attacking yields no benefit because that the defense completely nullifies the attacks.

So, for any $j \in C \setminus s_D$ with $w_j - c_{a,j} \leq 0$, including $j$ in $s_A$ provides non-positive utility while consuming budget. Thus we can restrict to $I^+ = \{i \in C \setminus s_D : w_i - c_{a,i} > 0\}$.

The attacker's problem becomes:

$$\max_{s_A \subseteq I^+} \sum_{i \in s_A} (w_i - c_{a,i})$$

$$\text{s.t.} \quad \sum_{i \in s_A} c_{a,i} \leq B_A$$

Define the knapsack instance $\mathcal{K} = (I^+, \{v_i\}_{i \in I^+}, \{c_{a,i}\}_{i \in I^+}, W)$:

- Item set: $I^+ = \{i \in C \setminus s_D : w_i - c_{a,i} > 0\}$ (profitable attacks)
- Item values: $v_i = w_i - c_{a,i} > 0$ for $i \in I^+$
- Item weights: $c_{a,i}$ for $i \in I^+$
- Knapsack capacity: $W = B_A$

Let's define the bijection $\phi : 2^{I^+} \leftrightarrow \{0,1\}^{|I^+|}$:

$$\phi(s_A) = x \text{ where } x_i = \begin{cases} 1 & \text{if } i \in s_A \\ 0 & \text{otherwise} \end{cases}$$

For any $s_A \subseteq I^+$ and $x = \phi(s_A)$:

$$\sum_{i \in s_A} (w_i - c_{a,i}) = \sum_{i \in I^+} (w_i - c_{a,i})x_i = \sum_{i \in I^+} v_i x_i \tag{3}$$

$$\sum_{i \in s_A} c_{a,i} \leq B_A \iff \sum_{i \in I^+} c_{a,i} x_i \leq B_A \tag{4}$$

Since $\phi$ is bijective and preserves both objective values and feasibility, the optimization problem becomes the corresponding knapsack problem:

$$\max_{x \in \{0,1\}^{|I^+|}} \sum_{i \in I^+} v_i x_i \tag{5}$$

$$\text{s.t.} \quad \sum_{i \in I^+} c_{a,i} x_i \leq B_A \tag{6}$$

### A.3.4 DEFENDER'S BILEVEL PROBLEM

The defender's optimization problem

$$\min_{x \in \{0,1\}^K} \quad f(x) = \sum_{i=1}^{K} c_{d,i} x_i + \mathcal{H}(x) \quad \text{s.t.} \quad \sum_{i=1}^{K} c_{d,i} x_i \leq B_D,$$

with attacker best-response

$$\mathcal{H}(x) = \max_{y \in \{0,1\}^K} \sum_{i=1}^{K}(w_i - c_{a,i})(1 - x_i)y_i \quad \text{s.t.} \sum_{i=1}^{K} c_{a,i}y_i \leq B_A,$$

is NP-hard.

*Proof.* We reduce from the 0/1 knapsack decision problem (KDP). An instance is $(\alpha_i, v_i, W, V)$ with item costs $\alpha_i > 0$, values $v_i \geq 0$, capacity $W$, and target $V$. The question is whether there exists $z \in \{0,1\}^K$ with

$$\sum_{i=1}^{K} \alpha_i z_i \leq W, \qquad \sum_{i=1}^{K} v_i z_i \geq V.$$

**Construction.** For each item $i$, create a client with

$$c_{d,i} = \alpha_i, \qquad c_{a,i} = 0, \qquad w_i = v_i.$$

Budgets are

$$B_D = W, \qquad B_A = 0.$$

Because $c_{a,i} = 0$ and $B_A = 0$, the attacker can always attack all unprotected clients. Thus, for any $x \in \{0,1\}^K$,

$$\mathcal{H}(x) = \sum_{i=1}^{K}(w_i - 0)(1 - x_i) = \sum_{i=1}^{K} v_i - \sum_{i=1}^{K} v_i x_i.$$

**Defender's objective.** The defender's cost is

$$f(x) = \sum_{i=1}^{K} c_{d,i}x_i + \mathcal{H}(x) = \sum_{i=1}^{K} \alpha_i x_i + \Big(\sum_{i=1}^{K} v_i - \sum_{i=1}^{K} v_i x_i\Big).$$

This simplifies to

$$f(x) = \Big(\sum_{i=1}^{K} v_i\Big) + \sum_{i=1}^{K}(\alpha_i - v_i)x_i,$$

subject to $\sum_i \alpha_i x_i \leq W$.

**Equivalence to knapsack.** Since $\sum_i v_i$ is constant, minimizing $f(x)$ is equivalent to maximizing

$$\sum_{i=1}^{K} v_i x_i \quad \text{s.t.} \sum_{i=1}^{K} \alpha_i x_i \leq W,$$

which is exactly the 0/1 knapsack optimization problem.

**Decision version.** Define threshold $L = \sum_i v_i - V$. Then

$$f(x) \leq L \quad \Longleftrightarrow \quad \sum_i v_i x_i \geq V,$$

under the same budget constraint $\sum_i \alpha_i x_i \leq W$. Thus solving the defender's decision problem solves KDP.

**Conclusion.** The reduction is polynomial-time, so the defender's problem is NP-hard. $\square$

### A.3.5 PRACTICAL ALGORITHMS AND IMPLEMENTATION

The NP-hardness result motivates the development of efficient algorithms that can handle realistic problem sizes while providing quality guarantees. We present three complementary approaches tailored to different system scales and computational constraints.

---

**Algorithm 1** Attacker Oracle

---

**Require:** Defense strategy $s_D$, attack budget $B_A$, damage weights $\{w_i\}$, attack costs $\{c_{a,i}\}$
**Ensure:** Optimal attack strategy $s_A^*$
 1: $I^+ \leftarrow \{i \in \mathcal{C} \setminus s_D : w_i - c_{a,i} > 0\}$ {Profitable targets}
 2: **if** $|I^+| \cdot B_A \leq \text{DP\_THRESHOLD}$ **then**
 3:    $s_A^* \leftarrow \text{DynamicProgramming}(I^+, \{w_i - c_{a,i}\}, \{c_{a,i}\}, B_A)$
 4: **else**
 5:    Sort $I^+$ by efficiency $\rho_i = (w_i - c_{a,i})/c_{a,i}$ in descending order
 6:    $s_A^* \leftarrow \emptyset$, budget $\leftarrow B_A$
 7:    **for** each $i \in I^+$ in sorted order **do**
 8:       **if** $c_{a,i} \leq$ budget **then**
 9:          $s_A^* \leftarrow s_A^* \cup \{i\}$
10:          budget $\leftarrow$ budget $- c_{a,i}$
11:       **end if**
12:    **end for**
13: **end if**
14: **return** $s_A^*$

---

**Attacker Oracle Implementation**   The attacker oracle is fundamental to all defender algorithms, as it computes the best response $R_A(s_D)$ for any given defense strategy $s_D$.

**Complexity Analysis:** The dynamic programming approach has complexity $O(|I^+| \cdot B_A)$, while the greedy approximation runs in $O(|I^+| \log |I^+|)$ time. The greedy algorithm achieves a $\frac{1}{2}$-approximation guarantee for the 0/1 knapsack problem.

**Defender Optimization Strategies**   We implement three defender algorithms with different computational-quality trade-offs:

**Exact Enumeration (Small Systems)**   For systems with $K \leq 20$ clients, we enumerate all feasible defense strategies:

---

**Algorithm 2** Exact Defender

---

**Require:** Client set $\mathcal{C}$, defense costs $\{c_{d,i}\}$, defense budget $B_D$
**Ensure:** Optimal defense strategy $s_D^*$
 1: $\mathcal{F}_D \leftarrow \{s_D \subseteq \mathcal{C} : \sum_{i \in s_D} c_{d,i} \leq B_D\}$ {Feasible defenses}
 2: $U^* \leftarrow -\infty, s_D^* \leftarrow \emptyset$
 3: **for** each $s_D \in \mathcal{F}_D$ **do**
 4:    $s_A \leftarrow \text{AttackerOracle}(s_D)$ {Algorithm 1}
 5:    $U \leftarrow -\sum_{i \in s_D} c_{d,i} - \sum_{i \in s_A \setminus s_D} w_i$
 6:    **if** $U > U^*$ **then**
 7:       $U^* \leftarrow U, s_D^* \leftarrow s_D$
 8:    **end if**
 9: **end for**
10: **return** $s_D^*$

---

**Complexity:** $O(2^K \cdot T_{\text{oracle}})$ where $T_{\text{oracle}}$ is the attacker oracle complexity.

**Greedy Heuristic (Medium Systems)**   For medium-sized systems, we use a protection efficiency heuristic:

**Complexity:** $O(K \log K)$ for sorting plus $O(K)$ for selection.

**Intuition:** The heuristic prioritizes clients with high damage-to-cost ratios, approximating the optimal trade-off between protection value and resource consumption.

**Bilevel Optimization (Large Systems)**   For large-scale problems, we use Sequential Least Squares Programming (SLSQP) with continuous relaxation:

---

**Algorithm 3** Greedy Defender

---

**Require:** Client set $\mathcal{C}$, damage weights $\{w_i\}$, defense costs $\{c_{d,i}\}$, budget $B_D$
**Ensure:** Defense strategy $s_D$
 1: Compute protection efficiency $\rho_i = w_i/c_{d,i}$ for all $i \in \mathcal{C}$
 2: Sort clients by $\rho_i$ in descending order: $\pi(1), \pi(2), \ldots, \pi(K)$
 3: $s_D \leftarrow \emptyset$, budget $\leftarrow B_D$
 4: **for** $j = 1$ to $K$ **do**
 5: $\quad i \leftarrow \pi(j)$
 6: $\quad$ **if** $c_{d,i} \leq$ budget **then**
 7: $\qquad s_D \leftarrow s_D \cup \{i\}$
 8: $\qquad$ budget $\leftarrow$ budget $- c_{d,i}$
 9: $\quad$ **end if**
10: **end for**
11: **return** $s_D$

---

**Algorithm 4** Bilevel SLSQP Defender

---

**Require:** Client set $\mathcal{C}$, parameters $\{w_i, c_{d,i}, c_{a,i}\}$, budgets $B_D, B_A$
**Ensure:** Defense strategy $s_D$
 1: Initialize $x_0 \in [0,1]^K$ {Relaxed defense variables}
 2: Define continuous attacker utility: $u_i(x_i) = (1 - x_i)(w_i - c_{a,i})$ for $i \notin s_D$
 3: Define $H_{\text{cont}}(x) = \max_{s_A}\{\sum_{i \in s_A} u_i(x_i) : \sum_{i \in s_A} c_{a,i} \leq B_A\}$
 4: Define objective $f(x) = \sum_i c_{d,i} x_i + H_{\text{cont}}(x)$
 5: Define constraint $g(x) = \sum_i c_{d,i} x_i - B_D \leq 0$
 6: $x^* \leftarrow$ SLSQP$(f, g, x_0)$ {Continuous optimization}
 7: $s_D \leftarrow \{i : x_i^* \geq 0.5\}$ {Rounding step}
 8: **if** $\sum_{i \in s_D} c_{d,i} > B_D$ **then**
 9: $\quad$ Sort $s_D$ by efficiency $w_i/c_{d,i}$ descending
10: $\quad$ Remove lowest-efficiency clients until $\sum_{i \in s_D} c_{d,i} \leq B_D$
11: **end if**
12: **return** $s_D$

---

**Implementation Details:**

- Fractional protection $x_i \in [0,1]$ reduces attacker utility by factor $(1 - x_i)$, interpolating between no protection ($x_i = 0$) and perfect protection ($x_i = 1$)
- SLSQP gradients computed via finite differences with step size $\epsilon = 10^{-6}$
- Convergence tolerance set to $10^{-6}$ for objective and constraint violations
- Maximum iterations limited to $100K$ to ensure termination

### A.3.6 ALGORITHM SELECTION STRATEGY

We automatically select algorithms based on problem characteristics:

$$
\text{Algorithm} = \begin{cases} \text{Exact} & \text{if } K \leq 20 \\ \text{Greedy} & \text{if } 20 < K \leq 100 \text{ and sparse budget} \\ \text{Bilevel} & \text{if } K > 100 \text{ or dense budget regime} \end{cases} \tag{7}
$$

where sparse budget refers to $B_D < \frac{1}{3} \min_i c_{d,i} \cdot K$ (can protect $< K/3$ clients) and dense budget allows protecting a significant fraction of clients.

### A.3.7 COMPUTATIONAL COMPLEXITY SUMMARY

**Scalability Analysis:** The framework scales to hundreds of clients through:

- Efficient attacker oracles with sub-millisecond query time for typical instances

| Algorithm | Time Complexity | Space | Approximation |
|-----------|-----------------|-------|---------------|
| Exact | $O(2^K \cdot |I^+| \cdot B_A)$ | $O(2^K)$ | Optimal |
| Greedy | $O(K \log K)$ | $O(K)$ | Heuristic |
| Bilevel | $O(T \cdot K \cdot |I^+| \log |I^+|)$ | $O(K)$ | Local optimum |

Table 2: Computational complexity comparison, where $T \leq 100K$ is the maximum SLSQP iterations.

- Parallelizable defense candidate evaluation in exact enumeration
- Sparse gradient computation in bilevel optimization exploiting problem structure
- Adaptive algorithm selection based on problem size and structure

### A.3.8  NUMERICAL STABILITY AND IMPLEMENTATION NOTES

**Finite Difference Gradients:** For the bilevel optimization, we compute gradients of $H_{\text{cont}}(x)$ using central differences:

$$\frac{\partial H_{\text{cont}}}{\partial x_i} \approx \frac{H_{\text{cont}}(x + \epsilon e_i) - H_{\text{cont}}(x - \epsilon e_i)}{2\epsilon}$$

**Tie-Breaking:** When multiple optimal solutions exist for the attacker's knapsack problem, we use lexicographic ordering based on client indices to ensure deterministic behavior.

**Budget Handling:** All algorithms include epsilon-tolerance ($\epsilon = 10^{-9}$) for budget constraint violations to handle floating-point precision issues.

## B  IMPLEMENTATION DETAILS

We implement FRL-SAGE in PyTorch on the `CartPole-v1` environment with $K = 25$ federated clients, each running local PPO and synchronizing a shared policy over repeated communication rounds. Client heterogeneity is introduced through tiered damage weights, while per-client defense and attack costs are drawn with strategic variety (scales in $[0.5, 1.5]$) and periodically reshuffled to induce non-stationarity. At the start of each round, the defender (leader) commits client-level protections under a fixed budget set to $5\%$ of the total defense cost, while the attacker (follower) adapts with a budget equal to $30\%$ of the total attack cost. We log training and validation returns alongside game-theoretic traces (number defended/attacked, theoretical damage, and realized performance drop), using a fixed seed for reproducibility.

## C  LLM USAGE DECLARATION

We acknowledge the use of large language models (LLMs) to assist in refining portions of the text in this manuscript. Their use was strictly limited to improving readability, grammar, spelling, and style. All ideas, interpretations, and conclusions presented in this work are solely the responsibility of the authors.

