# OpenReview forum: "FRL-SAGE: Stackelberg Game-Theoretic Defense Against Adaptive Adversaries in Federated Reinforcement Learning"
_ICLR.cc/2026/Conference — ICLR 2026 Conference Withdrawn Submission_

### Official Review · Reviewer_r5Qq · 2025-10-27

**Soundness:** 1
**Presentation:** 2
**Contribution:** 1
**Rating:** 2
**Confidence:** 4

**Summary:**

The paper proposes FRL-SAGE, a Stackelberg game framework where the defender commits to client-level protections and the attacker best-responds by selecting clients to compromise via gradient noise injection. The authors prove the attacker's problem reduces to knapsack while the defender's is NP-hard, then provide exact and approximate algorithms. Experiments on CartPole show marginal improvements, which the authors attribute to "deliberate volatility."

**Strengths:**

- The Stackelberg formulation with asymmetric utilities and budget constraints is mathematically clean
- Four adversarial scenarios (single/multi-client, static/dynamic) provide reasonable coverage
- Client heterogeneity via damage weights adds some realism over uniform importance assumptions

**Weaknesses:**

**Critical Blockers:**

1. **Failed Literature Review**. There is an established line of existing work on adversarial defense for Federated RL (Fault-Tolerant Federated Reinforcement Learning with Theoretical Guarantee, Fan et al., NeurIPS 2021; Decentralized Federated Policy Gradient with Byzantine Fault-Tolerance, Jordan et al., AAMAS 2024; Provably Robust Federated Reinforcement Learning, Fang et al., The Web Conf 2025. etc.). Most notably, Fan et al.'s NeurIPS 2021 paper provides the foundational work in this exact problem space. The paper cites Huang et al. 2021 for attacks on FRL but completely ignores the defense literature. How does the paper compare with the prior arts?

2. **Method inconsistency (REINFORCE vs PPO)**. The main text says clients “compute policy gradients using REINFORCE,” but the Implementation section later states “each running local PPO,” which is not a small detail and directly impacts reproducibility and results. The paper must be internally consistent or report both settings.

3. **Unrealistic defense/information assumptions**. Core theory and experiments assume perfect defense (protected clients cannot be attacked) and perfect information (attacker observes the defense set), which strongly favor the leader and may not hold in FRL practice; results risk over-estimating benefits. A probabilistic/partial protection model is deferred to future work, but the current conclusions depend on these assumptions.

4. **Damage weight mechanism lacks justification**. Client importance weights $w_i$ are central to the framework, directly scaling attack impact via $w_i/w_{ref}$ in gradient corruption. However:

   - No principled method provided for estimating $w_i$ in practice
   - Assumes defender knows attacker's valuation of clients (information asymmetry ignored)
   - Sensitivity analysis absent: how do misestimated $w_i$ affect equilibrium and performance?
   - Experiments use arbitrary tiered weights without justification

5. **No comparison to standard robust FL/FRL defenses**. Baselines are internal heuristics (“random / highest_risk / highest_value”), not established robust aggregators (Krum, Trimmed-Mean, Bulyan) or FRL-specific defenses, despite these being discussed in Related Work. This undermines the “state-of-the-art” wording.

**Significant issues:**

6. **Novelty claim needs scoping.** Framing FRL as a Stackelberg security game is fresh for FRL, but closely related Stackelberg formulations exist in supervised FL; novelty should be qualified as “FRL-specific” and not “first overall.” The admits this in Appendix A.1, but does not highlight it in the main body, which can be misleading.
7. **Only CartPole** - A 4-state toy problem cannot validate a security framework claiming applicability to "autonomous driving, healthcare, IoT control." This is insufficient for ICLR.
8. **Unconvincing results** - Even on CartPole, Figures 1-4 show minimal, inconsistent separation. Figure 2 shows baseline competitive with Stackelberg. No error bars or significance tests despite acknowledged high variance. When your method barely works on a toy problem under best-case assumptions, that's a red flag. And your view of "CartPole is deliberately volatile" needs more justification.

**Minor:**
9. Conclusion overstates findings given weak empirical results

**Questions:**

1. **Can you validate on standard FRL benchmarks?** CartPole is insufficient. Please evaluate on MuJoCo (HalfCheetah, Walker2d), Atari (Pong, Breakout), or multi-agent environments where FRL advantages are demonstrated.

2. **How do you estimate damage weights $w_i$ in practice?** What happens when they're misestimated by 50%? 100%? Can you provide robustness analysis?

3. **How does your framework compare to established Byzantine-robust aggregation?** Please compare against Krum, geometric median, trimmed mean applied to FRL.

4. **Can you extend to stronger attacks?** Gradient noise is weak. How does the framework perform against model replacement, backdoor, or adaptive attacks that learn defender's strategy?


5. **Can you relax perfect defense?** How does the framework generalize when protection reduces attack impact by (say) 70% rather than 100%?

6. **Statistical significance?** Please add paired t-tests or bootstrapped confidence intervals to demonstrate when Stackelberg significantly outperforms baselines.

7. **How does the framework scale?** Please evaluate with K ∈ {50, 100, 500} clients to demonstrate scalability claims.

---

### Official Review · Reviewer_CQRU · 2025-10-27

**Soundness:** 3
**Presentation:** 2
**Contribution:** 3
**Rating:** 2
**Confidence:** 4

**Summary:**

In this work, the authors use the federated reinforcement learning framework and setup a game theoretic problem with defender and attacker agents. They model it as a Stackelberg security game. The overall game involves a set of clients who are prone to attacks and both the defender and attacker have perfect knowledge of how much damage can be inflicted to different clients. They show that the attacker optimization can be solved as a 0/1 Knapsack problem and defender optimization is NP-Hard problem. They show experiments on CartPole domain.

**Strengths:**

The problem setup is novel in the paper and the overall topic is of interest to Game Theory, Federated RL subfields.

It is also interesting to see the reduction of attacker problem to 0/1 Knapsack problem and defender’s optimization as a pessimistic bi-level optimization.

Depending on the size of the number of clients, the authors analytically explore solutions for exact, greedy and approximation based strategies. The paper is analytically strong (though lacks in terms of easy to follow presentation)

**Weaknesses:**

The manuscript is not yet ready for publication. There are several sections which are hard to follow because of lack of details. I would recommend the authors to proofread the paper carefully — the current version requires major rewriting.

The first few sections for the manuscript are especially hard to make sense. Definitions are not precise and formal in nature and several times redefined with same variables for different terms.

The empirical experimentation lacks in terms of clarity and also in terms of comprehensibility. Different experimental setups are not explained. The paper directly goes into discussion of results. Additionally, just solving over one domain is not sufficient.

**Questions:**

The term gradient noise injection is referred to in several places in the paper and is not explained in detail.

SLSQP Is referred to without any introduction or formal discussion about it.

In the beginning, it is not clear how many types of agents there are — attender, defender, set of clients. It is not clear if the damage weights are associated with clients, attacker, or defender. It only becomes clear after the first half that there are three types of entities.

Definition 1 is redefined. Definition for FRL and FRL-SAGE uses same variable names.

“We adopt the general FRL framework introduced in Section 2,” is used while in Section 2 itself.

What is strategic importance of client? What is the intuition behind reference damage weight?

Damage weights are introduced after they are used in formulation

Line 228 unknown C symbol is used

---

### Official Review · Reviewer_9DLV · 2025-10-28

**Soundness:** 3
**Presentation:** 3
**Contribution:** 2
**Rating:** 2
**Confidence:** 3

**Summary:**

This paper explores the robustness of federated reinforcement learning under a set of adversarial clients. The authors formalize the problem as a Stackelberg game in which the defender allocates a limited budget to protect a group of clients while the adversary attempts to minimize performance by attacking the remaining clients. The authors prove the existence of an equilibrium and show that the attacker’s problem reduces to a 1/0 knapsack problem and that the defender’s problem is NP-hard. The authors further propose three simple algorithms and evaluate one of them in the CartPole environment.

**Strengths:**

1. The problem of robustness in Federated Reinforcement Learning with regard to faulty or adversarial clients is a relevant topic.
2. The assumptions are clearly formalized, and the proposed framework is modeled after a Stackelberg game.
3. Generally, the theoretical contributions, i.e. the proof of existence of the equilibrium, the reduction of the attacker’s problem, and the NP-hardness of the defenders problem are useful contributions to the community.

**Weaknesses:**

1. I like the theoretical contributions, but the empirical results seem unfinished. Currently, the experiments in the paper only consider one of the three settings, and therefore algorithms, that the authors proposed; the paper should, at the very least, provide one experiment for each setting.
2. Furthermore, many important aspects of the experiments are not explained. The baselines and some aspects of the experimental setup are never explained, which makes interpreting the results nearly impossible.
3. The experiments do not seem to show that the proposed method improves the robustness of the training.

While I appreciate the theoretical results of the paper, the experimental section, in its current form, is incomplete and poorly written. Overall, I believe the experimental section significantly detracts from the quality of the paper. Considering this paper as a theoretical contribution, I also believe the theoretical results need to be more novel since all the explored aspects of the problem seem straightforward.

**Questions:**

1. Can you define Single-client and Multi-client settings?
2. Can you define periodic shuffling?
3. How realistic is the assumption of perfect defense in a practical setting and on a high-level? This is not a criticism, as I believe it is a fair assumption as a starting point, I am just curious.
4. How realistic would a stochastic defense be in comparison, and, on a high-level, how would the setting change in case of such a defense?

---

### Official Review · Reviewer_1N1m · 2025-10-30

**Soundness:** 3
**Presentation:** 2
**Contribution:** 2
**Rating:** 4
**Confidence:** 4

**Summary:**

This paper presents FRL-SAGE, a Stackelberg game-theoretic defense framework designed for Federated Reinforcement Learning (FRL) under adaptive adversarial conditions. The authors argue that existing defenses in federated settings typically assume static adversaries, failing to model the dynamic strategic interactions between attackers and defenders over multiple training rounds. To address this gap, they cast the problem as a two-player Stackelberg security game where the defender (leader) allocates limited protection resources across clients while the attacker (follower) optimally selects clients to compromise within a budget. The paper formalizes this setup through asymmetric utility functions, proves the existence of Stackelberg equilibria, shows that the attacker’s optimization reduces to a 0/1 knapsack problem, and establishes NP-hardness for the defender’s bilevel optimization. To make the framework tractable, the authors propose exact, greedy, and relaxation-based approximation methods with theoretical guarantees. Experimental evaluations on the CartPole benchmark demonstrate that FRL-SAGE achieves modest but consistent improvements in robustness against adaptive gradient-noise injection attacks, supporting its potential as a foundational defense paradigm for adversarial FRL.

**Strengths:**

The primary strength of this work lies in its theoretical originality and formal rigor. The paper is among the first to explicitly formulate attacker–defender dynamics in federated reinforcement learning as a Stackelberg game, extending game-theoretic modeling from supervised federated learning to sequential policy optimization. The authors’ derivation of the attacker’s best response as a knapsack problem and the NP-hardness proof for the defender’s optimization are both mathematically elegant and logically sound. In terms of technical quality, the paper is well-grounded in theory, providing clear definitions, detailed proofs, and approximation analyses that ensure conceptual transparency. The introduction of client-specific damage weights to represent heterogeneous client importance adds a realistic dimension to the analysis. From a clarity standpoint, the writing is well-structured, notation is consistent, and assumptions are explicitly stated. In terms of significance, the paper’s contribution is to establish a principled foundation for robust and proactive defense in adversarial FRL, offering insights that can inform future work on adaptive and strategic robustness in distributed learning.

**Weaknesses:**

Despite its solid theoretical foundation, the paper’s empirical component is limited in both scope and depth. The experiments are restricted to the CartPole environment, which, while illustrative, is a simple and low-dimensional benchmark that does not sufficiently test the scalability or robustness of the proposed framework. The reported performance improvements are modest and sometimes inconsistent across attack scenarios, suggesting that further validation on more complex environments such as MuJoCo or multi-agent benchmarks would be beneficial. In addition, the comparison baselines focus primarily on heuristic strategies (random, highest-risk, highest-value) and do not include more recent or comparable game-theoretic and robust federated learning methods, such as FedGame or Meta-Stackelberg FL, limiting the interpretability of the reported gains. Some of the theoretical assumptions—such as perfect defense, perfect information, and full knowledge of client damage weights—are strong and may reduce the framework’s practical applicability. Finally, the experimental section would benefit from a more detailed analysis of computational overhead, convergence dynamics, and sensitivity to key parameters such as the attacker and defender budgets. Overall, while the theoretical contribution is sound, the empirical validation falls short of the standards expected for top-tier acceptance.

**Questions:**

Several questions could help clarify the contribution and potentially strengthen the work. First, how does FRL-SAGE scale when the number of clients grows significantly beyond the 25 used in experiments? An analysis of computational complexity or runtime behavior under larger system sizes would help demonstrate scalability. Second, could the framework be evaluated on more realistic or higher-dimensional RL environments to verify that the observed robustness trends generalize beyond CartPole? Third, how does FRL-SAGE compare against other recent game-theoretic or robust federated learning defenses such as FedGame (Jia et al., 2023) or Meta-Stackelberg FL (Li et al., 2024b)? Fourth, for the bilevel relaxation solved via SLSQP, do the continuous solutions approximate the discrete optima effectively, and can the authors quantify the approximation error empirically? Finally, given that FRL-SAGE assumes perfect knowledge of the clients’ damage weights, how feasible is this assumption in realistic federated learning deployments, and could the framework accommodate uncertainty or estimation noise in these values?

---

### Official Review · Reviewer_Spi5 · 2025-10-30

**Soundness:** 1
**Presentation:** 1
**Contribution:** 1
**Rating:** 0
**Confidence:** 5

**Summary:**

The paper considers an attacker-defender game in the context of federated RL. The paper presents an attacker-defender model based on prior work on attacker-defender interaction, adapted to the context of federated RL, and provides an algorithmic approach for computing a Stackelberg equilibrium. Numerical results are provided on a single environment.

**Strengths:**

The main strength of the paper is the alleged treatment of adversarial robustness in the context of federated reinforcement learning, as this is an area that has not received significant attention in the literature.

**Weaknesses:**

The main weakness of the paper is that the considered federated RL scenario is not well motivated, inconsistent with the attack model, the attack model itself is inconsistent with the motivation, the presentation is far from good, and the evaluation is insufficient. My detailed comments are as follows.

Authors argue in the introduction that "defenders also adapt covertly", i.e., defenders would adapt their strategy over time, in response to changes in the environment, in the training, etc, but this is not captures in the considered problem formulation.

The considered federated RL problem presented in Section 2.1 assumes that the MDPs faced by different clients may differ in their dynamics and the rewards. This assumption would imply that their policies would have to be different, making federated learning (without personalization) arguably a wrong fit for the problem. In terms of writing, the introduction states that "where poisoning often corrupts data or backdoors", but it is unclear why poisoning would corrupt backdoors in supervised FL.

In this very same subsection, the federated learning setup, including FedAvg is repeated, for a reason that is unclear.

In the sequel, authors provide a strategic game formulation of the security game, followed by a Stackelberg game formulation of the problem. It is unclear why there would be a need for the two formulations.

The threat model presented in Section 2.2 considers that client weights w_i are known to the attacker. It is unclear how these weights would be known prior to training. It is also somewhat unclear why the weights should be different, considering that authors assumes unweighted FedAvg in Section 2.

The threat model simplifies the problem of performing an adversarial attack deciding which clients to attack, as a static decision. It is unclear how a Byzantine attack would be coordinated, the gradient of which client would be optimal to manipulate in a certain gradient (and hence the updated model weights) and how. Overall, the threat model is rather simplistic in relation to the literature on adversarial attacks on (federated) learning, and appears to be a straightforward application of the security game framework. Aligned with this, most of the results (Knapsack formulation, etc) appear to be straightforward, the novelty compared to previous work is unclear.

The evaluation is based on a single environment. It is unclear from the paper (Appendix B) how the CartPole environments of the clients differed from each other (to match the RFL setup explainedin Section 2) in terms of dynamics and reward and how the weights were chosen (what is strategic variety, and what does it mean to scale in a closed interval). More details would be needed about the environment and the choice of the parameters, including the rationale for the choice.

**Questions:**

The paper argues that attackers are adaptive, but it is unclear how the considered Stackelberg game framework would take into account adaptation in the context of learning.

It is unclear what is the novelty of the security game framework (apart from the terms used in the paper, e.g., client instead of road or port) compared to previous work.

How would the client weights be determined in a practical setting a priori, without knowing the environments and how would the heterogeneity of the environments affect the results of the federated RL setup (and the efficiency of the attacks).

---

### Note · Authors · 2025-11-15

I have read and agree with the venue's withdrawal policy on behalf of myself and my co-authors.